# Identification of ligand and receptor interactions in CKD and MASH through the integration of single cell and spatial transcriptomics

**Jaime Moreno**[1], **Lise Lotte Gluud**[2,3], **Elisabeth D. Galsgaard**[4], **Henning Hvid**[5], **Gianluca Mazzoni**[1]*, **Vivek Das**[1]*

**1** Digital Science and Innovation, Computational Biology – AI & Digital Research, Novo Nordisk A/S, Maløv, Denmark, **2** Gastro Unit, Copenhagen University Hospital Hvidovre, Hvidovre, Denmark, **3** Dept of Clinical Medicine, University of Copenhagen, Copenhagen, Denmark, **4** Global Translation, Novo Nordisk A/S, Maløv, Denmark, **5** Global Drug Discovery, Novo Nordisk A/S, Maløv, Denmark

* gcma@novonordisk.com (GM); vvda@novonordisk.com (VD)

## Abstract

### Background

Chronic Kidney Disease (CKD) and Metabolic dysfunction-associated steatohepatitis (MASH) are metabolic fibroinflammatory diseases. Combining single-cell (scRNAseq) and spatial transcriptomics (ST) could give unprecedented molecular disease understanding at single-cell resolution. A more comprehensive analysis of the cell-specific ligand-receptor (L-R) interactions could provide pivotal information about signaling pathways in CKD and MASH. To achieve this, we created an integrative analysis framework in CKD and MASH from two available human cohorts.

### Results

The analytical framework identified L-R pairs involved in cellular crosstalk in CKD and MASH. Interactions between cell types identified using scRNAseq data were validated by checking the spatial co-presence using the ST data and the co-expression of the communicating targets. Multiple L-R protein pairs identified are known key players in CKD and MASH, while others are novel potential targets previously observed only in animal models.

### Conclusion

Our study highlights the importance of integrating different modalities of transcriptomic data for a better understanding of the molecular mechanisms. The combination of single-cell resolution from scRNAseq data, combined with tissue slide investigations and visualization of cell-cell interactions obtained through ST, paves the way for the identification of future potential therapeutic targets and developing effective therapies.

**Data Availability Statement:** The single-cell RNA sequencing (scRNAseq) liver data can be obtained from the public study conducted by Wang et al. in 2023. Additionally, all kidney data supporting the

findings of this study are available through The Kidney Precision Medicine Project (KPMP) at the following link: [https://www.kpmp.org/available-data]. The spatial liver data that support the findings of this study are not publicly available since it contains clinical information that could compromise the privacy of research participants. It is an observational prospective study and part of NCT04340817 which is currently ongoing where Novo Nordisk A/S has an ongoing early discovery research collaboration and generated the 10x Visium spatial transcriptomics data. Researchers interested in accessing the spatial liver data may make a reasonable request directly to the authors, or contact Datatilsynet via email at dt@datatilsynet.dk, or reach out to the Regional Ethics Committee in Denmark for Research (contact email: vek@regionh.dk). The source code used to analyze the data and produce the statistical figures is available on GitHub at [https://github.com/jaimomar99NN/L-R-spatial_scRNAseq-CKD-MASH].

**Funding:** This study was conducted in collaboration with Novo Nordisk A/S (Grant No. HvH161223), which provided research funding to Lise Lotte Gluud. None of the authors received salary funding for their contributions to this work. Novo Nordisk A/S supported the study by participating in study design, data collection, analysis, decision-making related to publication, and manuscript preparation in collaboration with the academic co-authors. Specific author roles are detailed in the 'Acknowledgements' section. There are no patents, products in development, or marketed products associated with this research. We confirm that these details do not alter adherence to PLOS ONE policies on sharing data and materials.

**Competing interests:** I have read the journal's policy and the authors of this manuscript have the following competing interests: VD, JM, GM, EDG and HH are employed by Novo Nordisk A/S. VD, JM, GM, HH and EDG hold minor stock portions as part of an employee-offering program. The authors have also indicated that no competing interests exist. Author Lise Lotte Gluud has received speaker honorarium from Norgine, Astra Zeneca, Sobi, Alexion and Novo Nordisk, consultant honorarium from Pfizer, Becton, Dickinson and Novo Nordisk and research funding from Alexion, Gilead Sciences and Novo Nordisk.

## 1. Background

Chronic kidney disease (CKD) and Metabolic dysfunction-associated steatotic liver disease (MASLD) are associated with obesity, type 2 diabetes, and metabolic syndrome [1, 2]. The possible association between MASH and CKD remains poorly understood [3, 4]. Both conditions are fibroinflammatory diseases, which involve a cycle where inflammation triggers fibrosis, and the resulting fibrosis can, in turn, perpetuate inflammation. A reduction of inflammation and prevention or slowing down of fibrosis could therefore be considered for MASLD as well as CKD [5–8]. Identification of common underlying alterations could provide essential information about disease development and progression.

MASLD is a condition characterized by hepatic steatosis and the presence of at least one cardiometabolic risk factor. The pathophysiology of MASLD is multifactorial with inflammation being a main driver of disease progression. In a subgroup of patients, MASLD can progress to metabolic dysfunction associated steatohepatitis (MASH), which is characterized by steatosis, inflammation, and ballooning [9]. MASH can lead to fibrosis development by triggering an excessive production of extracellular matrix, majorly by stellate cells, which is not adequately balanced by degradation [10].

Acute Kidney Injury (AKI) entails a potentially reversible loss of kidney function that can occur due to various causes, such as dehydration or infections [11]. Conversely, CKD is a progressive disease that results in irreversible damage to the kidneys and can lead to kidney failure. In certain cases, AKI can lead to CKD particularly if the injury is severe or recurrent. The development of both AKI and CKD is complex. Although severe inflammation, fibrosis, and lipid accumulation are well-established mechanisms involved in the development of kidney diseases, numerous other pathophysiological processes may be involved [12].

Various analytical methods have been developed to deduce cell-cell communication from the gene expression of individual cells using scRNAseq data [13]. These methods can infer the cellular communication between a signaling source cell (expressing the ligand) and the target cell (expressing the receptor) [14–16]. For this study, we utilized CellChat [17], a cell-to-cell communication tool that has emerged as one of the top-performing methods.

While distant cell interactions play a role in disease development, studying the molecular changes underlying paracrine signaling between adjacent cells may yield more promising results. This is likely to provide valuable information about the molecular changes underlying pathological lesions like development of fibrosis. In addition, inferring cell-cell communication based on L-R interactions solely based on single cells transcriptomics may lead to false positives due to lack of spatial localization of ligands and receptors in their respective cell types within a tissue. Spatial transcriptomics allows us to validate and understand these interactions and the tissue's cellular composition, enabling the comparison of gene expression between different cell types according to their locations and morphological features [18].

Currently, there are various spatial profiling methods, differing in resolution, number of genes profiled, area of the tissue captured, and technology used [19, 20]. Visium (from 10x Genomics) captures the expression of thousands of genes without predetermined gene targets, allows an unbiased assessment of gene expression changes. One challenge of the technique is to achieve the single cell resolution, as it depends on a grid of spots, each measuring 55 μm in diameter, with primers designed to assess the transcriptome profile of all cells (typically up to 10–15 cells) located in each spot. A solution is to use a deconvolution technique based on the expression within each spot, which dissects the mixed signals from individual cells within a spot, providing an estimation of the abundance of the cell types in the spot using a scRNAseq dataset as a reference [21]. This information helps to uncover the spatial organization of different cell types and their interactions within the tissue.

Previous studies have combined scRNAseq and ST data, [22–27] but none have integrated disease conditions with possible overlapping pathophysiology. Through our analysis, we uncover changes in gene expression and cell types involved in the two fibroinflammatory conditions CKD and MASLD. Our work has the potential to subsequently facilitate drug discovery efforts by pinpointing targets and enhancing our understanding of the disease progression.

## 2. Materials and methods

### Data collection for CKD and AKI

The kidney data used in this study is from the Kidney Precision Medicine Project (KPMP) and a file per sample was obtained from the KPMP cohort (S1 Table) [28, 29]. The single nuclei dataset consists of 29 samples from 13 healthy individuals, 6 patients with AKI, and 10 patients with CKD. The single nuclei dataset includes 16 different cell types. The spatial dataset comprises 15 10X Visium fresh frozen samples, including 6 with AKI and 9 with CKD (S1 Table).

### Data collection for MASLD

The hepatic single nuclei dataset analysis was carried out using a publicly available dataset [30], that consists of 12 samples: 3 healthy individuals and 9 patients with MASH (7 F1-F2 fibrosis and 2 F3-F4 fibrosis). These 9 patients with MASH were combined in our analysis (S2 Table).

The spatial transcriptomics data from liver samples were obtained from the Copenhagen Cohort (Coco) of MASLD, which is a prospective cohort study (Clinicaltrials.gov NCT04340817, H-17029039) that aims to systematically evaluate biomarkers and potential drug targets in patients with MASLD and MASH performed at the Gastro Unit, Copenhagen University Hospital Hvidovre. The healthy participants and patients underwent clinical assessment, routine blood tests, a Fibroscan, and a liver biopsy, which was percutaneous or transjugular (healthy controls only underwent transjugular biopsies). All biopsies were evaluated by two expert hepatopathologists based on steatosis, ballooning, inflammation, and fibrosis [31].

The spatial transcriptomics was available for 24 samples, including 6 healthy individuals and 18 patients with MASLD (9 with F1-F2 fibrosis and 9 with F3-F4 fibrosis). Two samples (with F4 fibrosis) that did not fulfil the criteria for MASH were excluded from the analyses. From FFPE liver biopsies, sections of 5 μm thickness were cut and mounted onto positively charged Visium slides (one sample mounted in each capture area of 6.5 x 6.5 mm) and processed for spatial transcriptomics according to the 10X Genomics Visium FFPE Version 1 protocol. Briefly, samples were deparaffinized, stained with hematoxylin and eosin (H&E) and imaged using VS200 Slide Scanner (Olympus Life Science) prior to decrosslinking, destaining and overnight probe hybridization with the 10X Visium Human version 1 probe set. The following day, hybridized probes were released from the tissue, and ligated to spatially barcoded oligonucleotides on the Visium Gene expression slide. Barcoded ligation products were then amplified, and a library was constructed from this pre-amplified sample. Libraries from all 24 samples were pooled and sequenced on a NovaSeq 6000 sequencing platform (Illumina), using a NovaSeq 6000 S2 Reagent Kit v1.5 (Illumina) according to the manufacturer instructions. Subsequently, fastq files were generated for each sample, reads were aligned to their corresponding probe-sequences (Visium human transcriptome probe set v1, based on GRCh38 2020-A) and mapped back to the Visium spot where a given probe were originally captured, and finally aligned to the original H&E-stained image of the tissue section, using Space Ranger version 1.3.0 (10X Genomics). From the Space Ranger output, the Loupe browser file for each sample was used for initial inspection of data, and the filtered count matrix was used for further downstream processing and analysis of data.

## Quality control, integration, and cluster annotation

For consistency in the project, both diseases were filtered using the same thresholds for cells or spots, depending on the technology employed. These thresholds included a minimum of 500 counts, a minimum of 250 expressed genes, a mitochondrial percentage of less than 20%, and a log10 ratio of the number of genes over the number of counts of at least 0.80. Basic quality control metrics can be observed in S1 and S2 Figs. The kidney single nuclei data was already processed and annotated, consisting of more than 169,000 cells (S3 Fig). In contrast, the liver single nuclei data underwent a downstream analysis, consisting of normalization, dimensionality reduction, and clustering where the Seurat functions were employed [32]. In addition, the cell clusters were annotated based on well-known markers (S5 Table) and by examining the differentially expressed gene markers obtained per cluster in the PanglaoDB [33]. It yielded over 205k cells, which were classified into eight distinct cell types (S4 Fig). Subsequently, the spatial data was normalized and then integrated using Harmony [34], with default settings, removing the batch size per sample and per diseases status. The liver dataset yielded almost 26,000 spots, while the kidney dataset integrated over 8,400 spots (S5 Fig).

## Deconvolution analysis

To perform the deconvolution process, we used the Bayesian model Cell2location [35]. It takes the gene expression signature of the cell types in the scRNAseq data to estimate the abundance of each cell type at each spot. The model was trained with the scRNAseq data for 500 epochs with a batch size of 2500. Afterwards, the intersection of the genes between the two technologies were identified and used for the spatial mapping. Finally, the regression model was set up for deconvolution with hyperparameters priors: expected cells per spot (n = 8) and detection sensitivity ($\alpha = 20$). The final model was trained for 20,000 epochs. Quality control plots of the deconvolution along with comparison can be seen in S6 Fig.

## Co-occurrence and compositional clusters

We performed co-occurrence analyses to investigate non-random associations between cell types within the tissue. To this end, we utilized the ISCHIA method, a computational combinatorics framework that assigns a quantitative property to the interaction potential of cell types by computing their spatial co-occurrence [36]. This method enables us to identify pairs of cell types that exhibit positive co-occurrence in specific cellular neighborhoods, indicating that the observed co-occurrence is higher than expected by chance. To obtain specific cellular neighborhoods, the data was partitioned into clusters of similar cellular composition. To achieve this, the abundance of cell types per spot matrix was subjected to a k-means clustering, and the total Within-Cluster Sum of Squares was calculated for a range of one to 15 clusters. The optimal k value per disease was subsequently selected by the elbow method (S7 Fig). By examining the cell types across the composition clusters and the number of spots contributing to each disease status, two composition clusters were selected for each disease. Finally, the co-occurrence of each cluster was calculated.

## Co-expression in the tissue

In this study, various new functions were deployed to visualize the co-expression of multiple genes in the ST data. These functions utilize a color-coding scheme to represent the co-expression value, enabling more precise visualization of specific regions of the tissue and facilitating comparison between different sets of genes. The threshold parameter can be employed to remove noise and enhance the accuracy of the visualization. The co-expression is based on the

minimum principle, which stipulates that the expression of the genes is at least the minimum of them. In addition, the spatial correlation was performed to validate the co-expression of the ligand and receptor pairs with markers driving the diseases. Hence, the Pearson correlation between the normalized expression level of these markers per slide was computed considering all the samples or divided by disease state [37]. The markers chosen from literature were validated by pathologists.

## Cell-cell communication and L-R interactions

CellChat [17], was utilized to study the cell-cell interactions per disease status. Each scRNAseq processed dataset was divided according to the state of the disease and interactions with a p-value below 0.05 were considered significant. CellChat was run with default parameters. Additionally, the CellChat function *identifyOverExpressedGenes* was applied to conduct a differential expression analysis of ligands and receptors pairs. The natural log fold-change threshold was set to 0.1 with an adjusted P value threshold set to 0.05 for both ligand and receptor. Only ligands and receptors pairs with the same directionality in the regulation were taken. In the kidney, we calculated Healthy vs AKI, Healthy vs CKD, and AKI vs CKD, while in the liver, we computed Healthy vs F1-F2, Healthy vs F3-F4, F1-F2 vs F3-F4 and finally Healthy vs all samples with MASH.

## Integration of methodologies

To harness the maximum information of both technologies, a comprehensive analytical workflow was developed combining the methods described. Firstly, the cell-cell communication and L-R interactions were calculated in the single cell data. From those, only the cell types that are positively co-occurring in spatial transcriptomics were taken. Differential expression analysis was then performed, taking the signaling that was either up or down regulated in the disease. Finally, L-R pairs that passed all these criteria were mapped onto the slides to explore the co-expression between them. A visual overview of the different workflows can be seen in Fig 1.

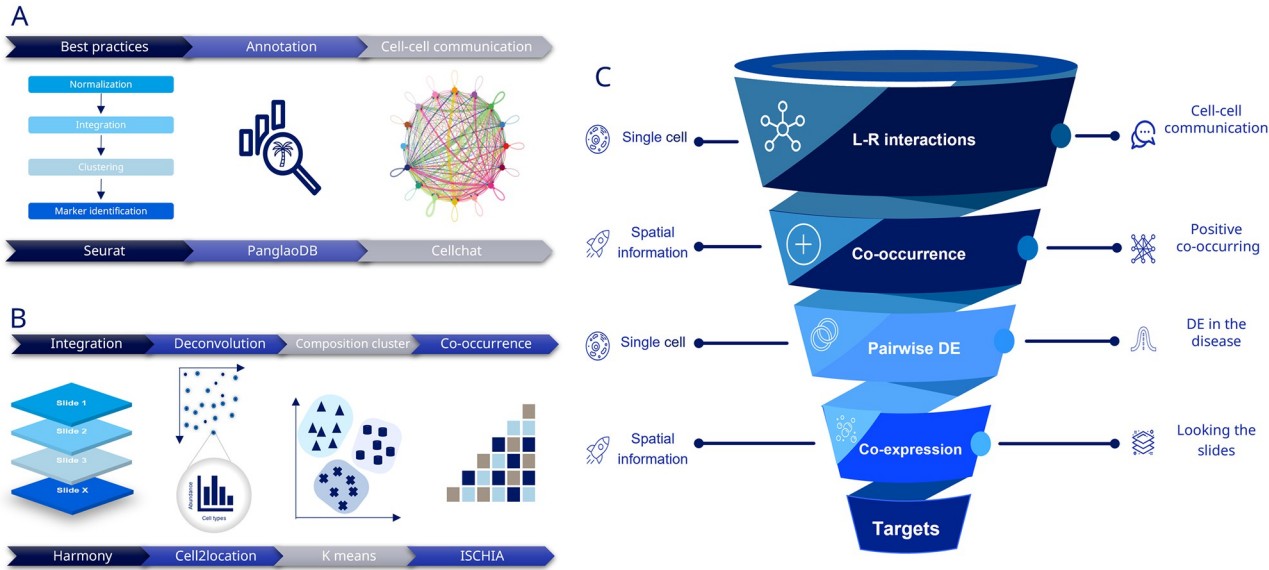

**Fig 1. Methods workflow. A**. Single cell pipeline. **B**. Spatial transcriptomics pipeline. **C**. Combined pipeline for target discovery.

## 3. Results

We studied common cell-cell communication within each tissue across the spectrum of the disease progression. For the kidney dataset, we compared healthy, AKI, and CKD patients. For the dataset focusing on MASLD, we compared samples from healthy subjects, subjects with mild/moderate (F1-F2) fibrosis, and subjects with advanced (F3-F4) fibrosis and well as healthy versus all subjects with MASH. Lastly, we focused on finding shared cell-cell interactions in both diseases, potentially revealing common therapeutic targets.

### a. Kidney spatial deconvolution and scRNAseq identifies major disease signal in the communication between proximal tubule and immune cells

In kidney, we clustered the spatial dataset based on the deconvolution results, revealing six different compositional clusters (CC) (Fig 2A). CC2 and CC6, were selected for further analysis based on presence of disease relevant cell types and the presence across disease status (Fig 2B). CC2, CC3 and CC4 were enriched in proximal tubule (PT), fibroblasts (FIB) and immune cells (IMM). Among them, CC2 was chosen because of having a higher number of spots in CKD and AKI, CC6 was selected due to its strong presence in CKD samples and its enrichment in IMM and FIB, but not PT. To study the spatial co-localization of different cell types, co-occurrence was calculated for selected clusters resulting Fig 2C.

Subsequently, we analyzed cell-cell communication to identify L-R interactions on the scRNAseq data kidney biopsies. Of all the cell-cell communications obtained, only the cell types positively co-occurring in the two CCs selected were taken for further analysis.

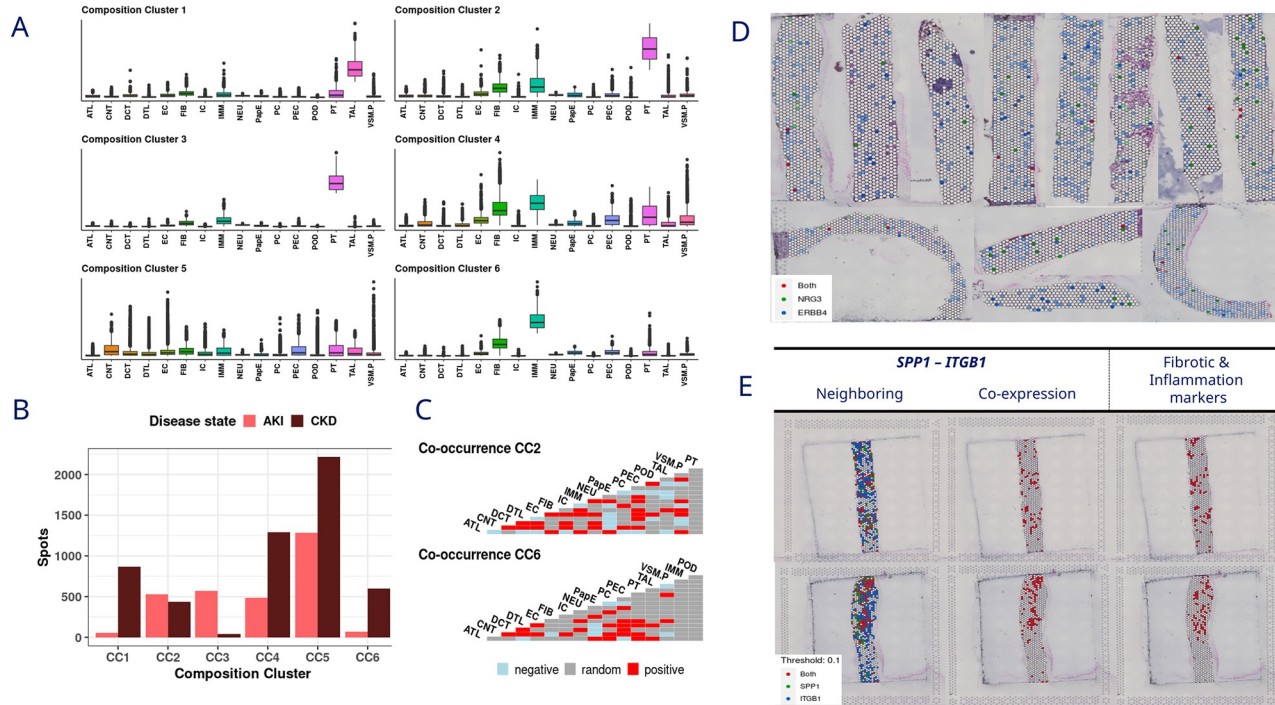

**Fig 2. Kidney results. A**. Composition clusters of the cell deconvolution results in spatial data. **B**. Contribution of the spots to each disease status per composition cluster. **C**. Cell type co-occurrence within the composition clusters two and six. **D**. Co-expression in the spatial data of the *NRG3-ERBB4* interaction (*NRG3*, green; *ERBB4*, blue; both, red). **E**. Co-expression in the spatial data of the *SPP1-ITGB1* interaction (*SPP1*, green; *ITGB1*, blue; both, red). Fibrotic markers and inflammation markers: *COL3A1* and *TGFB1* (red).

**Table 1. Number of L-R interaction differentially regulated across the different kidney disease status.**

| L-R Pairwise DE | Number of interactions | |
|---|---|---|
| | Up regulated | Down regulated |
| **H vs AKI** | 85 | 49 |
| **H vs CKD** | 21 | 131 |
| **AKI vs CKD** | 12 | 159 |

Furthermore, we identified up and down regulated L-R pairs (Table 1) by conducting a differential expression (DE) analysis between the kidney disease status (See cell-cell communication section in Methods 5).

**i. Integrative analysis reveals increased neuregulin 3 (*NRG3*)- Receptor tyrosine-protein kinase erbB-4 *(ERBB4)* signaling and crosstalk between tubular and glomerular cell types in AKI and CKD.** The next step in our analysis was to confirm the L-R interactions identified in scRNAseq by looking at its co-expression within same areas of the tissue slides from kidney biopsies.

The L-R interactions showing consistent association with the disease were investigated. The ligand *VEGFA* and its receptor *VEGFR1* were found to be consistently down regulated in the disease tissue samples. Conversely, *NRG3* with its receptor *ERBB4* were up regulated in AKI and CKD. Interestingly, PT cells expressing *NRG3* were communicating with descending thin limb cells, endothelial cells and the PT themselves expressing *ERBB4* showing an upregulation in CKD compared to both the healthy and AKI states (Table 2).

To investigate whether co-expression of L-R pairs occurs in the same tissue region, we visualized co-expression of the L-R pair within the tissue slide. The co-expression of *NRG3* and *ERBB4* could not be validated since the expression of the ligand (*NRG3*) was detected only in a few spots (Fig 2D). Since *NGR3* was found to have high expression in scRNAseq data (S10 Fig), the reason for its low detection in spatial transcriptomics is due to limitations in the technology.

**ii. The comparison between healthy and CKD revealed metabolic reprogramming associated with the interaction of *SPP1* and the receptor complex (*ITGAV-ITGB1*).** We focused our analysis on the L-R pairs differentially regulated between H vs CKD. We validated

**Table 2. Cell communication between the *NRG3-ERBB4* interaction in kidney.**

| L-R Pairwise DE | Source | Target | Ligand logFC | Receptor logFC |
|---|---|---|---|---|
| **H vs AKI** | EC | CNT | 0.30 | 0.10 |
| | PT | CNT | 0.10 | 0.10 |
| | EC | DCT | 0.30 | 0.27 |
| | EC | IC | 0.30 | 0.15 |
| | EC | POD | 0.30 | 0.25 |
| | POD | POD | 0.34 | 0.15 |
| **H vs CKD** | PT | DTL | 0.24 | 0.20 |
| | PT | EC | 0.24 | 0.65 |
| | PT | PT | 0.24 | 0.69 |
| **AKI vs CKD** | PT | DTL | 0.14 | 0.66 |
| | PT | EC | 0.14 | 0.35 |
| | PT | PT | 0.14 | 0.58 |

NT: Connecting tubule, DCT: Distal convoluted tubule, DTL: Descending thin limb, EC: Endothelial, IC: Intercalated cells, POD: Podocytes, PT: Proximal tubular cells.

the L-R pairs by computing the spatial co-expression patterns within the tissue. Interestingly, *SPP1* and the receptor complex *ITGAV–ITGB1*, exhibited either up regulation or down regulation depending on the cell types involved in the communication (S6 Table). While the expression of this complex receptor is upregulated in fibroblasts from CKD samples, it is consistently downregulated in endothelial cells. The spatial co-expression of *SPP1* and *ITGAV1-ITGB1* can be observed by plotting the spatial neighboring co-expression of the L-R pair (Fig 2E). Moreover, the involvement of the L-R pairs in the disease progression can be appreciated by visualizing the spatial co-expression of the L-R pair with fibrotic and inflammation markers (*COL3A1* and *TGFB1*) [38].

## b. Liver spatial deconvolution and scRNAseq cell-cell communication identifies major disease signal in stellate and endothelial cells

In liver samples, two of the CCs identified (CC1 and CC4) (Fig 3A) were selected based on the number of spots contributing to the liver fibrosis stages (Fig 3B). In the selected CCs, all cell types are positively co-occurring except for hepatocytes (Fig 3C).

Following the same approach as in kidney, we calculated the cell-cell communication in the scRNAseq dataset, focusing exclusively on L-R interactions between cell types that exhibited positive co-occurrence in the spatial analysis. The L-R DE analysis between different fibrosis stages resulted in Table 3.

**i. Down regulation of shared communication is identified in MASH compared to healthy and in advanced fibrosis compared to mild/moderate fibrosis.** Among all pairwise

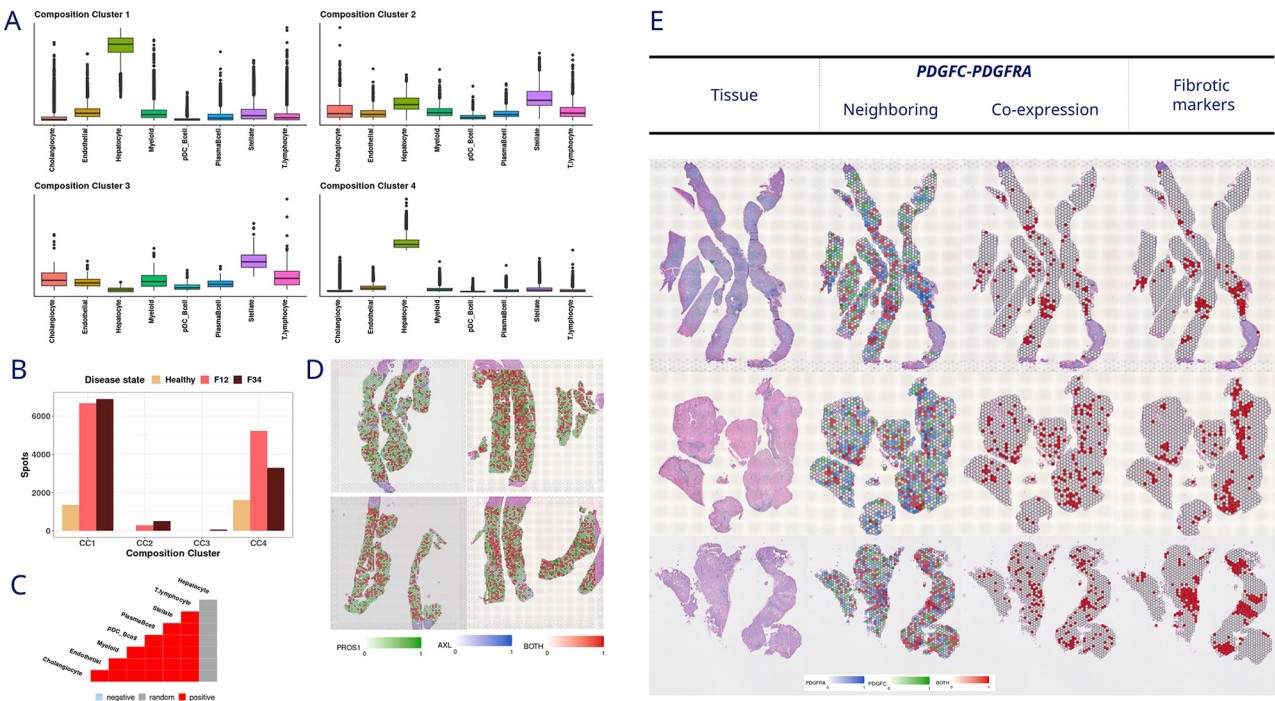

**Fig 3. Liver results. A**. Composition clusters of the cell deconvolution results in spatial data. **B**. Contribution of the spots to each disease status per composition cluster. **C**. Co-occurrence of the cell types in composition clusters one and four, same co-occurrence. **D**. Co-expression in the spatial data of the PROS1-AXL interaction (*PROS1*, green; *AXL*, blue; both, red). **E**. Co-expression in the spatial data of the *PDGFC-PDGFRA* interaction (*PDGFC*, blue; *PDGFRA*, blue; both, red. Tissue: H&E stain. Fibrotic markers: *LUM*, *IGFBP7* and *COL1A1* (red).

**Table 3. Number of differential interactions across the conditions in liver.**

| L-R Pairwise DE | Number of interactions | |
|---|---|---|
| | Up regulated | Down regulated |
| **H vs F1-F2** | 10 | 19 |
| **H vs F3-F4** | 38 | 24 |
| **F1-F2 vs F3-F4** | 47 | 16 |
| **H vs MASH** | 13 | 16 |

comparisons analyzed, we focused on those whose expression regulation was consistent across all comparisons. Three L-R pairs (*BMP6-(BMPR1A-BMPR2), BMP6-(BMPR1B-BMPR2) and PROS1-AXL)* were down regulated in advanced fibrosis and potentially involved in the communication between endothelial cells and stellate cells and in the autocrine signaling in myeloid cells (Table 4). The expression of *BMP6* decreases in endothelial cells, along with the expression of receptor complexes *BMPR1A-BMPR2* and *BMPR1B-BMPR2* in stellate cells. Similarly, the receptor *PROS1* and the ligand *AXL* exhibit the same pattern both in myeloid cells. Subsequently, interaction between *PROS1* and *AXL* was visually confirmed by evaluating the co-expression on the slides (Fig 3D). Unfortunately, *BMP6* is not part of the FFPE probe.

**ii. Comparing the healthy versus the MASH state leads to the identification of up regulation of *PDFGC-PDGFRA* which overlaps with fibrotic areas.** In liver, we focused on the L-R associated with MASH. The interaction between the ligand *PDGFC* and the receptor *PDGFRA* in the communication between cholangiocytes with stellate cells, and myeloid with stellate cells, implies that the interaction by this L-R pair, and hence the communication between these cell types, intensifies in MASH.

In liver, as the disease progresses, areas with fibrosis and dense lymphocytic infiltration become visible in the H&E histology sections appearing as dark blue regions in the low-magnification tissue images in Fig 3E [39, 40]. It allows us to visually observe if the co-expression of L-R interactions overlaps with those areas, as well as fibrotic markers in liver [41], namely *LUM, IGFBP7* and *COL1A1*. In addition, Pearson correlation between L-R and fibrotic markers was used to quantify the colocalization with fibrotic areas (S9 Fig), seeing an increment in the NASH samples compared with the healthy samples.

## c. Common L-R interactions in CKD and MASH support endothelial cells expressing INSR playing a critical role in cardio metabolic diseases

In liver, the communication between cholangiocytes, myeloid and stellate cells expressing *NAMPT* with endothelial cells expressing *INSR* was up regulated in MASH subjects compared to healthy ones. Contrarily, in kidney, the communication between fibroblasts, parietal epithelial and distal tubule cells with endothelial cells expressing *NAMPT* and *INSR*, respectively, was found to be down regulated in the CKD state versus the healthy status. A summary describing the communication and expression in both tissues can be found in Table 5.

**Table 4. Common cell communication across all L-R pairwise DE comparisons in liver.**

| Source | Target | Interaction |
|---|---|---|
| Endothelial | Stellate | *BMP6-(BMPR1A-BMPR2)* |
| Endothelial | Stellate | *BMP6-(BMPR1B-BMPR2)* |
| Myeloid | Myeloid | *PROS1-AXL* |

**Table 5. Cell communication between the *NAMPT-INSR* interaction comparing healthy versus disease samples in both tissues.**

| Tissue | Source | Target | Ligand logFC | Receptor logFC |
|---|---|---|---|---|
| Liver | cholangiocytes | EC | 24.1 | 95.2 |
| | myeloid | EC | 25.4 | 95.2 |
| | stellate | EC | 39.6 | 95.2 |
| Kidney | FIB | EC | -0.8 | -0.39 |
| | PEC | EC | -1.1 | -0.39 |
| | DTL | EC | -0.55 | -0.39 |

Source: cell type expressing the ligand, Target: cell type expressing the receptor, FIB: fibroblasts, EC: endothelial, PEC: parietal epithelial, DTL: descending thin limb cells.

In both liver and kidney, the immune cells expressing *CD44* were down regulated in the disease. The communication between cholangiocytes expressing *SPP1* and *CD44*-positive T cells was down regulated in MASH. Likewise, the ascending thin limb cells expressing the *SPP1* were down regulated in immune cells expressing *CD44* in CKD.

## 4. Discussion

In recent years, the understanding of the pathophysiology underlying CKD and MASH has advanced significantly, thanks to the development of high throughput technologies such as single-cell and spatial omics. In this work we uncovered ligand-receptor complexes that emerges in MASH and CKD driving communications between cell types that are key player in the fibro-inflammatory and metabolic processes including stellate and cholangiocytes cells in the liver and proximal tubular cells in kidney.

In kidney, we identified L-R interactions that exhibited consistent regulation in the expression in the diseased subject, *NRG3-ERBB4* were up regulated in the disease, while *VEGFA--VEGFR1* were downregulated along the progression of the disease. The cell-cell communication analysis identified *NRG3-ERBB4* as key players in the communication between proximal tubules with descending thin limb cells and with endothelial cells. This suggests that upon reaching the chronic state, PT cells increase the *NRG3* expression, intensifying their communication with the cell types that are also augmenting the expression of *ERBB4*. Consistent with the findings in the literature, PT plays a key role in CKD and an increased activity in the PT can lead to an increased risk of CKD development [42, 43].

Although the literature does not provide direct evidence of the association of *NRG3* and renal fibrosis or disease progression, numerous publications have reported the involvement of *ERBB* in various renal diseases [44]. *ERBB4* is a member of the epidermal growth factor receptor family, a group of genes that has been shown to exhibit increased activity in AKI and CKD [45, 46]. Notably, *ERBB4-IR*, a novel long non-coding RNA located on chromosome 1 in the mouse genome [47], has been implicated in mediating renal fibrosis [48, 49]. Moreover, there is evidence that *ERBB4-IR* plays a role in CKD [47, 50] and diabetic nephropathy [51, 52]. While these studies support the importance of *ERBB4* in renal diseases in mice, limited human results are available to support these findings [53]. Our data suggests that expression of *ERBB4* is increased in human CKD and plays a role in the pathogenesis by increased paracrine signaling between proximal tubules, descending thin limb cells and endothelial cells. The expression of *NRG3* and *ERBB4* genes was not consistently detected in the spatial transcriptomics dataset due to the low expression of the ligand *NRG3* within the tissue which contrasts with its expression observed in the single nuclei dataset.

In the comparison of healthy versus CKD samples, we identified 150+ interactions highlighting the complexity of the mechanisms underlying the disease (Table 1). Particularly, the *SPP1-(ITGAV-ITGB1)* interaction exhibits distinct patterns: the receptor is consistently expressed by fibroblasts in upregulated communications and by endothelial cells in downregulated communications in disease setting. This specificity in cell types associated with communication changes offers exciting prospects for precise and effective target discovery in the context of the disease.

The L-R pair *SPP1-(ITGAV-ITGB1)* is clearly co-expressed and overlaps with fibrotic and inflammation markers, *COL3A1* and *TGFB1*. However, when the expression pattern of a L-R pair varies between cell types (up or down regulated), interpretation of the biological role is complicated and challenges to see the overlapping with the markers selected. The involvement of this interaction, *SPP1-(ITGAV-ITGB1)*, has been extensively studied in the literature. *SPP1* is known to impact not only AKI and CKD [54–56] but also several cardiovascular diseases [57–59]. Likewise, *ITGA9* and *ITGB1*, genes belonging to the integrin family are also recognized for their role in CKD [60–62]. However, their role in disease progression remains unclear, primarily due to the involvement of multiple cell types and processes. In this study, we have improved the understanding of how the *SPP1* ligand interacts with complex integrins to mediate fibrosis in fibroblasts and immune cell types where they are up regulated while in endothelial cells and distal or connecting tubule, they are down regulated. This metabolic reprogramming may be attributed to various post-transcriptional and post-translational events within the realm of regulatory biology that occurs during disease process which is out of scope in this current study. Hence, further studies with other types of omics profiling beyond mRNA analysis are required for a deeper understanding of the diverse cell-type specific complex regulatory biology.

In liver, we observed co-occurrence of all cell-types except hepatocytes in the chosen compositional cluster. Hepatocytes represent the most abundant cell type in liver and represent 70–80% of the liver [63]. Even if there is no statistically significant enrichment between hepatocytes and other cell-types, we assume hepatocytes are physically in contact with all the other cells in the liver. However, in this analysis, we focused on non-hepatocytes such as hepatic stellate cells, which are known for being key drivers of liver fibrosis [64].

Three L-R pairs interactions were found consistently down regulated in the disease, two of them between endothelial cells expressing the ligand *BMP6*, and stellate cells expressing either the complex receptor *BMPR1A-BMPR2* or *BMPR1B-BMPR2* [65], and the other interaction between the myeloid cells with themselves expressing the ligand *PROS1* and the receptor *AXL*. These results suggest a decreased communication between the cell types with the progression of the disease. Unfortunately, *BMP6* is not captured by the probes of 10X Genomics Visium FFPE and therefore, the co-expression of the interaction in the ST dataset cannot be evaluated, needing further studies to validate the results.

Our work identifies *BMP6* as a key ligand in liver fibrosis in humans. Interestingly, *BMP6* has been previously identified as a target in murine models where the enhancement of its expression inhibits hepatic fibrosis in liver disease [66].

Finally, our findings highlight a potential autocrine signaling of myeloid cells between the ligand *PROS1* and the receptor *AXL* that diminishes with the progression of the disease. The neighboring expression of *PROS1* and *AXL* were observed across all tissue sections. These findings are in good agreement with previous studies which explored the role of AXL in liver injury and fibrosis [67–71].

Remarkably, our study revealed that the *PDGFC-PDGFRA* interaction served as one of the L-R pairs responsible for the communication between cholangiocytes and stellate cells, as well as myeloid with stellate cells. Notably, the expression of this interaction was found to be

up regulated in MASH compared to healthy samples. Hence, we observed not only co-expression between the interaction but also a significant overlap with fibrotic areas within the tissue and the co-expression of fibrotic markers. Additionally, the correlation between the ligand and receptor with these markers revealed strong spatial correlation. The co-localization of *PDGFC- PDGFRA* interaction with fibrotic areas and markers was notably more pronounced in slides obtained from MASH patients with fibrosis stage 3 and 4 scores, compared to healthy samples or those with fibrosis stage 1 and 2 scores. This distinction is also observed in the spatial correlation calculated by condition status, where an increase in the Pearson correlation coefficient is seen comparing MASH samples versus healthy subjects. Consequently, our findings provide novel evidence suggesting that an increased interaction of *PDGFC* and *PDGFRA* between stellate cells with cholangiocytes, as well as myeloid cells, is associated with liver disease progression and enhanced fibrotic damage. The *PDGFC* gene, a member of the platelet-derived growth factor family and its association with liver diseases has been broadly studied [72]. Remarkably, the association of *PDGFC* expression has been linked not only to the progression of fibrosis, but also to chronic inflammation, increment of collagen production, hepatocarcinogenesis, steatosis, hepatocellular carcinoma [73–79]. Moreover, the expression of *PDGFC* in hepatic stellate cells, as observed in our study, has been previously reported [80].

Interestingly, common interactions were also found in both diseases with shared cell types involved. While *NAMPT-INSR* was up regulated in liver, it was down regulated in kidney. However, the cell type expressing the receptor in both cases were the endothelial cells. Previous studies have corroborated the decreased *INSR* expression in kidney diseases [81, 82], while others have validated the increase of the receptor in liver diseases, highlighting the involvement of stellate cells in the process [83]. In addition, research has pinpointed the association between *NAMPT* in obesity and diabetes [84–87], as well as its function in MASH [88, 89], which highlights one more time the complex mechanisms within these metabolic diseases.

Another L-R pair *(SPP1-CD44)* was downregulated in CKD and MASH. The receptor, *CD44*, was found on T-cells in the liver and on immune cells in the kidney, suggesting a correlation between the two diseases. *SPP1* has already been identified as a relevant ligand playing a significant role in kidney disease in our study. Additionally, our results indicate that SPP1 has a key role in liver disease, where it communicates with T cells expressing *CD44*.

Our findings are in line with the complexity of these two multifactorial diseases characterized by multiple ligand-receptor pairs whose changes in abundance and physiological effect are dependent on the cell type expressing them. Our analysis provides a better disease understanding by identifying dysregulated L-R pairs and changes in the communication between different cell types during disease progression and holds significant potential for improving target identification and validation.

Lastly, this study has certain limitations. Quality control of both scRNAseq and ST datasets are crucial since the cell-cell communication and L-R pairs obtained from the scRNAseq dataset and validated within the spatial information. While the kidney dataset presents a pairwise arrangement, where both datasets are from the same cohort, the scRNAseq and ST datasets in liver come from entirely distinct studies. Hence potential batch effects and variations in experimental conditions need to be considered while analyzing the deconvolution outcome and interpreting the overall results.

Finally, further studies such as in situ hybridization or functional studies experiments may be necessary to confirm the functional roles of identified genes with their interactions and identify potential therapeutic targets.

## 5. Conclusion

Integration of single-cell and spatial transcriptomics provides advantages that can help overcome limitations of individual technologies. The findings of our study contribute to a deeper comprehension of the molecular mechanisms behind complex diseases.

In this study, we developed a novel approach to utilize all the information provided by the deconvolution analysis, improving the accuracy of the results, and providing more detailed insights into target discovery and validation. Our framework identifies differentially expressed genes and elucidates potential ligand-receptor pairs and associated cell types involved in communication, offering a more nuanced understanding of CKD and MASH.

Our analytical framework identified mechanisms and cell-cell signaling as evidenced in the literature, while others may be potentially considered as novel discoveries. It is also noteworthy that few of the literature validated ligand-receptor interactions are from studies in mice, while our study utilizes human data. This approach provides a strong human-centric and translational relevance around candidate drug target (potential ligand-receptor interaction) behavior and their cell-cell communication pattern during disease process.

Overall, this study paves a way for gaining a better understanding of complex diseases and identifying potentially new therapeutic targets via integrating scRNAseq and ST modalities.

## Supporting information

**S1 Table. Number of samples per condition for both datasets used in kidney.**
(CSV)

**S2 Table. Number of samples in liver.** *Two samples did not fulfill the criteria for MASH.
(CSV)

**S3 Table. Number of disease samples with diabetes history for each dataset.**
(CSV)

**S4 Table. Number of disease samples per Estimated Glomerular Filtration Rate (eGFR) in the kidney datasets.**
(CSV)

**S5 Table. Markers used for annotation of the scRNAseq liver.**
(CSV)

**S6 Table. SPP1–(ITGA9-ITGB1) cell signaling in H vs CKD.** CNT: Connecting tubule, DCT: Distal convoluted tubule, DTL: Descending thin limb, FIB: Fibroblasts, IMM: Immune, TAL: Thick ascending limbs.
(CSV)

**S1 Fig. Quality control of the scRNAseq data.**
(TIF)

**S2 Fig. Quality control of the ST data.**
(TIF)

**S3 Fig. Number of nuclei per cell type per disease status in the kidney scRNAseq dataset.**
(TIF)

**S4 Fig. Number of nuclei per cell type per disease status in the liver scRNAseq dataset.**
(TIF)

**S5 Fig. Visualization of the integrated embeddings of the spatial slides of the two tissues colored by disease status.**
(TIF)

**S6 Fig. Cell2location quality control metrics of both tissues compared with the official vignette.** A. 2D histogram showing the reconstruction accuracy. B. ELBO loss history during training deconvoluting the spatial datasets.
(TIF)

**S7 Fig. Elbow plot of the compositional cluster selection after the deconvolution process.**
(TIF)

**S8 Fig. Liver results considering the MASH score.** A. Composition clusters of the cell deconvolution results in spatial data. B. Contribution of the spots to each disease status per composition cluster.
(TIF)

**S9 Fig. Pearson spatial correlation of the *PDGFC-PDGFRA* interaction with fibrotic markers in the liver dataset.** A. Correlation considering all the samples. B. Correlation considering only Healthy samples. C. Correlation considering only MASH samples.
(TIF)

**S10 Fig. Expression of *NRG3* and *ERBB4* in the kidney scRNAseq dataset.**
(TIF)

## Acknowledgments

GM and VD conceived the study. JM prepared the data and ran the Bioinformatics formal analysis. JM prepared the plots and the initial draft of the manuscript under the supervision of GM and VD. HH generated the liver spatial transcriptomics data. JM, GM, and VD interpreted the results. EDG, LL and HH provided additional interpretation of the results. All authors read, edited, and reviewed the final manuscript. We thank Thomas Monfeuga and Cesar Augusto Prada Medina for providing the Liver Single Nuclei data with annotated cell types as well as Atefeh Lafzi and Albin Gustav Sandelin for the guidance and support of the project.

## Author Contributions

**Conceptualization:** Gianluca Mazzoni, Vivek Das.

**Data curation:** Jaime Moreno.

**Formal analysis:** Jaime Moreno.

**Funding acquisition:** Gianluca Mazzoni, Vivek Das.

**Investigation:** Jaime Moreno, Gianluca Mazzoni, Vivek Das.

**Methodology:** Jaime Moreno.

**Project administration:** Gianluca Mazzoni, Vivek Das.

**Resources:** Henning Hvid, Vivek Das.

**Software:** Jaime Moreno.

**Supervision:** Gianluca Mazzoni, Vivek Das.

**Validation:** Lise Lotte Gluud, Elisabeth D. Galsgaard, Henning Hvid, Gianluca Mazzoni, Vivek Das.

**Visualization:** Jaime Moreno, Gianluca Mazzoni.

**Writing – original draft:** Jaime Moreno, Gianluca Mazzoni.

**Writing – review & editing:** Lise Lotte Gluud, Elisabeth D. Galsgaard, Henning Hvid, Gianluca Mazzoni, Vivek Das.

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
