## [Decision Letter · Decision Letter 0]

30 Jan 2024

PONE-D-23-42927Identification of ligand and receptor interactions in CKD and MASH through the integration of single cell and spatial transcriptomicsPLOS ONE

Dear Dr. DAS,

Thank you for submitting your manuscript to PLOS ONE. After careful consideration, we feel that it has merit but does not fully meet PLOS ONE’s publication criteria as it currently stands. Therefore, we invite you to submit a revised version of the manuscript that addresses the points raised during the review process.

Moreno et al studied ligand-receptor interactions in two metabolic fibroinflammatory diseases (ie, CKD and MASH) by integrated analysis of data from single-cell RNA-seq and spatial transcriptomics. They generated the spatial transcriptome of human NASH samples, which will greatly benefit the research in this field.

To make it suitable for publication, please clarify all the concerns raised by the reviewers and revise the manuscript accordingly.

In addition, the authors used liver scRNA-seq data from Ref 30 and wrote that “The hepatic single nuclei dataset analysis was carried out using a publicly available dataset (30), that consists of 18 samples: 3 healthy individuals, 15 patients with MASH”. However, there were 3 healthy human, 9 patients with MASH and 1 mouse MASH in Ref 30 (GSE212837). Please double-check the resource and verify the information.

We look forward to receiving your revised manuscript.

Kind regards,

Xianmin Zhu

Academic Editor

PLOS ONE

“I have read the journal's policy and the authors of this manuscript have the following competing interests: VD, JM, GM, EDG and HH are employed by Novo Nordisk A/S. VD, JM, GM, HH and EDG hold minor stock portions as part of an employee-offering program. The authors have also indicated that no competing interests exist. Author Lise Lotte Gluud has received speaker honorarium from Norgine, Astra Zeneca, Sobi, Alexion and Novo Nordisk, consultant honorarium from Pfizer, Becton, Dickinson and Novo Nordisk and research funding from Alexion, Gilead Sciences and Novo Nordisk.”

3. For studies involving third-party data, we encourage authors to share any data specific to their analyses that they can legally distribute. PLOS recognizes, however, that authors may be using third-party data they do not have the rights to share. When third-party data cannot be publicly shared, authors must provide all information necessary for interested researchers to apply to gain access to the data. (https://journals.plos.org/plosone/s/data-availability#loc-acceptable-data-access-restrictions)

a) A description of the data set and the third-party source

b) If applicable, verification of permission to use the data set

c) Confirmation of whether the authors received any special privileges in accessing the data that other researchers would not have

d) All necessary contact information others would need to apply to gain access to the data

4. We notice that your supplementary figures are uploaded with the file type 'Figure'. Please amend the file type to 'Supporting Information'. Please ensure that each Supporting Information file has a legend listed in the manuscript after the references list.

Reviewers' comments:

Reviewer's Responses to Questions

**Comments to the Author**

1. Is the manuscript technically sound, and do the data support the conclusions?

Reviewer #1: Partly

Reviewer #2: Yes

2. Has the statistical analysis been performed appropriately and rigorously? 

Reviewer #1: Yes

Reviewer #2: Yes

3. Have the authors made all data underlying the findings in their manuscript fully available?

Reviewer #1: Yes

Reviewer #2: Yes

4. Is the manuscript presented in an intelligible fashion and written in standard English?

Reviewer #1: Yes

Reviewer #2: Yes

5. Review Comments to the Author

Reviewer #1: The manuscript presents an integration framework to identify ligand and receptor interactions in CKD and MASH. My major concerns are the following:

1. How reliable when using Seurat to annotate the spots?

2. The spatial data should revel some morphology features among the spots. How does the framework use such features to help identify the LR interactions?

3. can the author illustrate the deconvolution results

4. When the data are having dropout issues, can the proposed framework enhance the gene expression and reveal the latent correlations

5. Since the ST data have location information, why not consider the graph information?

Reviewer #2: The authors aimed to identify novel disease target by study ligand-receptor pairs in both CKD and MASH. They developed a bioinformatics pipeline to identify ligand-receptor pairs associated with disease by integrating and cross-checking spatial transcriptomics and single nuclear RNA-seq data. It’s a nice try even though authors didn’t further validate their findings with further experiment. Their study and open source code will inspire more audience to mine scRNA-seq and ST data more comprehensively, and potentially enhance drug target discovering through cutting-edge technologies.

6. PLOS authors have the option to publish the peer review history of their article (what does this mean?). If published, this will include your full peer review and any attached files.

Reviewer #1: No

Reviewer #2: No

---

## [Author Response · Author response to Decision Letter 0]

8 Mar 2024

Response to Reviewers 

Journal requirements

1. Ensure manuscript meets PLOS ONE requirements.

The requirements have been addressed in the Manuscript.

2. Competing interests

“I have read the journal's policy and the authors of this manuscript have the following competing interests: VD, JM, GM, EDG and HH are employed by Novo Nordisk A/S. VD, JM, GM, HH and EDG hold minor stock portions as part of an employee-offering program. The authors have also indicated that no competing interests exist. Author Lise Lotte Gluud has received speaker honorarium from Norgine, Astra Zeneca, Sobi, Alexion and Novo Nordisk, consultant honorarium from Pfizer, Becton, Dickinson and Novo Nordisk and research funding from Alexion, Gilead Sciences and Novo Nordisk.”

The statement has been added to the Cover letter and the manuscript (lines 487 and 488 in the tracked changes manuscript)

3. Third-party data 

a) A description of the data set and the third-party source

b) If applicable, verification of permission to use the data set

c) Confirmation of whether the authors received any special privileges in accessing the data that other researchers would not have

d) All necessary contact information others would need to apply to gain access to the data

The scRNAseq liver data can be obtained from the public study: Wang S, et al. 2023. (30) and GSE212837. All the kidney data that supports the findings of the study can be obtained in The Kidney Precision Medicine Project (KPMP) [https://www.kpmp.org/available-data]. However, the spatial liver data that support the findings of this study are not publicly available. It is an observational prospective study and part of NCT04340817 which is currently ongoing where Novo Nordisk A/S has an ongoing early discovery research collaboration and generated the 10x Visium spatial transcriptomics data. It contains clinical information that could compromise the privacy of research participants, hence can be made available from the authors upon reasonable request and approval from The Danish Data Protection Agency. 

4. Change supplementary figures 

Supplementary figures are uploaded with the file type 'Figure'. Please amend the file type to 'Supporting Information'. Please ensure that each Supporting Information file has a legend listed in the manuscript after the references list.

Names of the supplemental has been fixed.

Journal concern 

“In addition, the authors used liver scRNA-seq data from Ref 30 and wrote that “The hepatic single nuclei dataset analysis was carried out using a publicly available dataset (30), that consists of 18 samples: 3 healthy individuals, 15 patients with MASH”. However, there were 3 healthy human, 9 patients with MASH and 1 mouse MASH in Ref 30 (GSE212837). Please double-check the resource and verify the information “

Thanks for highlighting this issue. The overall patient number have been corrected in the revised manuscript. Lines 99 and 100 in the tracked changes manuscript reflects the revised numbers.

Comments from the reviewers

1. How reliable when using Seurat to annotate the spots?

In this study we have used cell2location instead of Seurat to deduce the cell type proportions within the spots which is a standard deconvolution method used to annotate spots in spot-based Spatial transcriptomics. It is a well benchmarked probabilistic-based and one of the most accurate and top-ranking methods as per B Li et al. 2022, Yan et al. 2023, and Sangaram C et al 2023. Seurat is not used to annotate 10x Visium ST data due to the spot resolution limitation, where expression is averaged among adjacent cells, introducing significant noise for clustering. Additionally, in the case of the MASH data, there is usually a large proportion of hepatocytes in liver. This makes clustering and manual annotation based on known markers impractical since hepatocytes are present in all spots. Finally, it is also important to note that cell types that are minimally identified and expressed in the tissue would be impossible to annotate using the traditional approaches in 10X Visium technology. For these reasons, we chose to estimate the cell type proportions within the spots using the expression of a reference single-cell dataset using a probabilistic model (Cell2location). 

2. The spatial data should reveal some morphology features among the spots. How does the framework use such features to help identify the LR interactions?

Our framework leverages morphological features to validate the targets identified in the scRNASeq dataset. For instance, in the liver, we examined whether the expression of the L-R overlapped with fibrotic areas visible in the tissue morphology and well-known fibrotic markers. In the case of the kidney, the dataset is fresh frozen, which compromises the quality of morphology, making it more challenging to utilize features among the spots. Nevertheless, we compared the L-R found with the expression of inflammation markers in the kidney.

Furthermore, pathologists validated the morphological features to check their correlation with these targets.

It is important to clarify that this project does not involve any classical image analysis. However, we do utilize image information by overlaying the expression with fibrotic markers that are identifiable in the tissue, and we quantify their similarity by calculating the Pearson correlation (Figure S9) .

3. Can the author illustrate the deconvolution results?

The deconvolution results are presented in FigS6 and are also compared with the main vignette of cell2location to assess the quality of the deconvolution. All the details around the model training and hyperparameter information are available under sub-section Deconvolution analysis under Section 2. Materials and Methods. Additionally, we are providing the images below that represent two samples from the data used, one from kidney tissue and another from liver tissue demonstrating the cell type proportion per spot in a slide.

4. When the data are having dropout issues, can the proposed framework enhance the gene expression and reveal the latent correlations?

Thank you for this insightful question. In scRNAseq, dropout events typically occur due to technical limitations preventing RNA detection. In the case of 10X Visium data, the spot resolution averages gene expression across multiple cells, which can potentially mitigate dropout events. However, in the framework we have developed, targets are initially identified in the scRNAseq data and subsequently validated using spatial data. Consequently, if a dropout event occurs in scRNAseq, it would not be identified as a target, and thus, currently our framework would not detect it.

We are currently exploring an alternative approach where targets are identified and selected from the spatial data and compared with those in single-cell data. However, to ensure robustness and accuracy in such, we need better spatial technologies with near single cell resolution which is a limitation in 10x Visium ST. We plan to address such with advanced platforms like Xenium or Visium HD to refine this framework at Novo Nordisk once available.

5. Since the ST data have location information, why not consider the graph information?

 Thank you for your insightful question. Indeed, we have considered the idea of integrating graph information to investigate Ligand-Receptor (L-R) interactions within the spatial data. However, due to the absence of single-cell spatial resolution, this approach could potentially introduce a significant amount of noise. This is particularly true when considering the inclusion of cell types in the graph, as hepatocytes are ubiquitously present in all spots in the Visium ST data. There are existing methods, such as CLARIFY (Zhang X, et al., 2023), which create graphs at the gene and cell levels. However, these methods are only used with technologies that provide single-cell spatial resolution due to the aforementioned limitation.

In our research, the main objective is to address the constraints associated with spot-based ST technology. Therefore, we focus on the abundance of cell types within the spots and cluster them based on their proportions. This approach allows us to identify tissue zones with similar cellular compositions. Moreover, we corroborate the cell types identified in scRNAseq by performing a statistical co-occurrence analysis. This analysis helps us ascertain whether the cell types involved in each compositional cluster are positively co-occurring. Such co-occurrence suggests a potential functional role in those specific tissue areas where they co-locate.

---

## [Decision Letter · Decision Letter 1]

1 Apr 2024

PONE-D-23-42927R1Identification of ligand and receptor interactions in CKD and MASH through the integration of single cell and spatial transcriptomicsPLOS ONE

Dear Dr. DAS,

Thank you for submitting your manuscript to PLOS ONE. After careful consideration, we feel that it has merit but does not fully meet PLOS ONE’s publication criteria as it currently stands. Therefore, we invite you to submit a revised version of the manuscript that addresses the points raised during the review process.

 Please make some minor revisions on the Materials and Methods and supplemental tables. In line 96, there should be total 12 samples (3 healthy individuals and 9 patients with MASH). In the Table S2, the numbers should be corrected accordingly. Specifically, "F1-F2" should be "7" instead of "13".  "MESH" total of F1-F2 and F3-F4 should be "9" instead of "15".  The title of the table indicates" 1 Two samples did not fulfill the criteria for MASH". So please label "1" in the table. In the Table S3, "Unknown" "ScRNAseq" should be "9" instead of "15".  Btw, please carefully double check the text because, for example, in line 619, Ref 41 shows "Invalid citation". 

We look forward to receiving your revised manuscript.

Kind regards,

Xianmin Zhu

Academic Editor

PLOS ONE

Journal Requirements:

Reviewers' comments:

Reviewer's Responses to Questions

**Comments to the Author**

1. If the authors have adequately addressed your comments raised in a previous round of review and you feel that this manuscript is now acceptable for publication, you may indicate that here to bypass the “Comments to the Author” section, enter your conflict of interest statement in the “Confidential to Editor” section, and submit your "Accept" recommendation.

Reviewer #1: All comments have been addressed

Reviewer #2: All comments have been addressed

2. Is the manuscript technically sound, and do the data support the conclusions?

Reviewer #1: Yes

Reviewer #2: Yes

3. Has the statistical analysis been performed appropriately and rigorously? 

Reviewer #1: Yes

Reviewer #2: Yes

4. Have the authors made all data underlying the findings in their manuscript fully available?

Reviewer #1: Yes

Reviewer #2: Yes

5. Is the manuscript presented in an intelligible fashion and written in standard English?

Reviewer #1: Yes

Reviewer #2: Yes

6. Review Comments to the Author

Reviewer #1: Thanks for responding to my previous comments. The revision has addressed my concerns. I think it is acceptable.

Reviewer #2: The authors' answers to reviewers' questions and suggestions are reasonable and sound. I would suggest accepting this paper for publication.

7. PLOS authors have the option to publish the peer review history of their article (what does this mean?). If published, this will include your full peer review and any attached files.

Reviewer #1: **Yes: **ziyang tang

Reviewer #2: No

---

## [Author Response · Author response to Decision Letter 1]

8 Apr 2024

Response to Reviewers

Journal requirements. 

In line 96, there should be total 12 samples (3 healthy individuals and 9 patients with MASH).

Number of samples are now fixed.

In the Table S2, the numbers should be corrected accordingly. Specifically, "F1-F2" should be "7" instead of "13". "MESH" total of F1-F2 and F3-F4 should be "9" instead of "15". The title of the table indicates" 1 Two samples did not fulfill the criteria for MASH". So please label "1" in the table.

Number of samples in the Table S2 has been updated. Label “1” has been replaced with “*” to ease formatting in the excel file.

In the Table S3, "Unknown" "ScRNAseq" should be "9" instead of "15". 

Table S2 has been now fixed.

Please review your reference list to ensure that it is complete and correct. If you have cited papers that have been retracted, please include the rationale for doing so in the manuscript text, or remove these references and replace them with relevant current references. Any changes to the reference list should be mentioned in the rebuttal letter that accompanies your revised manuscript. If you need to cite a retracted article, indicate the article’s retracted status in the References list and include a citation and full reference for the retraction notice. Line 619, Ref 41 shows "Invalid citation".

A problem with Endnote was causing the invalid citation, it has now been fixed.

---

## [Editor Report · Decision Letter 2]

10 Apr 2024

Identification of ligand and receptor interactions in CKD and MASH through the integration of single cell and spatial transcriptomics

PONE-D-23-42927R2

Dear Dr. DAS,

We’re pleased to inform you that your manuscript has been judged scientifically suitable for publication and will be formally accepted for publication once it meets all outstanding technical requirements.

Kind regards,

Xianmin Zhu

Academic Editor

PLOS ONE
---

## [Editor Report · Acceptance letter]

8 May 2024

PONE-D-23-42927R2 

PLOS ONE

Dear Dr. DAS, 

I'm pleased to inform you that your manuscript has been deemed suitable for publication in PLOS ONE. Congratulations! Your manuscript is now being handed over to our production team.

Kind regards, 

on behalf of

Dr. Xianmin Zhu 

Academic Editor

PLOS ONE